# Physical and Mechanical Properties of Reclaimed Asphalt Pavement (RAP) Incorporated into Unbound Pavement Layers

Christina Plati *, Maria Tsakoumaki and Konstantinos Gkyrtis

Laboratory of Pavement Engineering, School of Civil Engineering, National Technical University of Athens, GR-15773 Athens, Greece
* Correspondence: cplati@central.ntua.gr

**Abstract:** Against the backdrop of global warming and depletion of natural resources, new techniques and alternative materials need to be explored and integrated into road construction. Reclaimed Asphalt Pavement (RAP) is one of the waste materials that can be reused in new road projects if its behavior is better understood. Numerous researchers have studied the use of RAP in both bound and unbound pavement layers. However, the mechanical behavior and deformation characteristics of RAP in unbound pavement layers are not fully understood due to its unique properties. For this reason, this paper aims to investigate the performance of RAP in the construction of unbound pavement layers (base and subbase). The methodology used consists of two phases: (i) laboratory tests in terms of physical properties, bearing capacity and permanent deformations generated and (ii) a comparative analysis of the test results. For the laboratory tests, the RAP material was taken from the milling operation of a pavement section to be rehabilitated and blended with virgin aggregates (VA) in different proportions. In addition, a sample consisting of pure VA was used as a reference sample for the comparative analysis of the results. Overall, it is concluded that the use of RAP for admixture in unbound layers is feasible and meets the sustainability requirements of pavement materials and structures without compromising pavement strength. A highlight of the research findings is that RAP with percentages up to 40% is a rational approach for the development of RAP-VA mixes to be incorporated into unbound pavement layers. Nevertheless, the results of the present study support the statement that testing is required each time to define the capabilities of RAP considering local effects and material conditions.

**Keywords:** pavement; unbound layers; RAP; CBR; resilient modulus; permanent deformation

## 1. Introduction

In the context of environmental protection and more sustainable development, new techniques and alternative materials must be researched and integrated into construction processes. Road projects, in particular, require large quantities of conventional materials for new construction and/or rehabilitation of pavements, further depleting already limited natural resources [1]. This has led road engineers to explore alternative materials that are currently considered "waste materials" [2]. Therefore, the ability to recycle and use these materials can help reduce the need to obtain and process virgin aggregates for road projects. This, in turn, has environmental benefits, such as reducing energy consumption and greenhouse gas (GHG) emissions during the production and construction process and avoiding the landfilling of construction and demolition (C&D) materials. This also results in financial benefits, such as lower costs for the procurement of materials for the construction of pavement layers [3,4]. On the other hand, C&D materials can only partially replace conventional materials for road construction due to their relatively lower strength [5]. This is a significant issue, as the response of materials in underlying layers, namely base/subbase and subgrade layers, under various conditions is important for the stability of numerous pavement projects [6]. There are plenty of studies dealing with the technical properties of

C&D materials for road construction, both for surface courses [3,7] and for base courses and substructures [8,9].

Moving forward, Reclaimed Asphalt Pavement (RAP) is one of the waste materials that can be rationally reused in new road projects, if its behavior is better understood. RAP is obtained by milling or removing the asphalt layers from road pavements that are either being maintained or rehabilitated. It consists of natural aggregates (93–97% by weight of mixture) covered with aged bitumen and the remaining cured asphalt (3–7%) [10]. RAP did not become of interest to the road industry until the 1970s, due to the petroleum crisis, when its potential for reincorporation into asphalt mixtures to produce new asphalt layers became better understood. From that point on, knowledge about the incorporation of RAP into asphalt layers grew steadily, and today there are standardized testing protocols for asphalt layers treated with RAP [11,12]. This is also confirmed by Jaawani et al. [10], who reported that nearly 70% of recent studies on RAP involve its use in asphalt layers. However, as higher percentages of RAP were included in asphalt mixtures, variations were observed in these mixtures, resulting in its use generally being limited to only 15% [13]. This restriction on the use of RAP in asphalt layers has led to an increase in stockpiles of RAP worldwide. In the USA, for example, only one-third of the RAP material is reused per year, while this ratio is one-sixth in Europe [14].

Because of the widespread availability of this potentially sustainable material, research has begun to take a new direction in evaluating the potential of RAP for incorporation into unbound base courses or subbases of road pavements [15]. The suitability of RAP material for use in unbound pavement layers has been systematically investigated by several researchers [16–19]. However, the mechanical responses and deformation characteristics of the RAP material in unbound pavement layers are not yet fully known due to its special properties. Specifically, although it is widely accepted that the RAP-VA blends have a comparable or better bearing capacity than pure VA [16–19], other properties are rather questionable. Many studies [16,20,21] report that CBR values decrease when RAP is added to VA, but other studies [22] report the opposite result. Additionally, regarding the permanent deformation of RAP-VA blends, there are studies which claim either a significant development [23–25] or an insignificant development of permanent deformations compared to pure VA [26–28]. As far as the physical properties of RAP-VA blends are concerned, the optimum moisture content (OMC) and maximum dry density (MDD) have a wide range of values, which are comparable or better than pure VA [22,25,28–30].

Consequently, it seems that there is still room for further research in order to investigate and evaluate RAP as a sustainable material for the construction and/or rehabilitation of unbound pavement layers (base course and subbase). Therefore, the objective of this study is to evaluate the RAP material based on laboratory procedures, in terms of its physical and mechanical properties, including bearing capacity and permanent deformations generated. The results are discussed in comparison with the properties of Virgin Aggregates (VA), with the ultimate aim of determining an optimum ratio of VA and RAP in a blend used for the construction of unbound pavement layers.

## 2. Review of Related Studies

The main concern of the research community is to determine the optimal dosage of RAP in new mixtures without compromising pavement performance. Since RAP is investigated in this study as a component of the unbound layers, it is important to evaluate not only the physical properties, but also (i) the bearing capacity and (ii) the permanent deformations that may cause rutting in the pavement. Therefore, a brief literature review of the existing evidence on these two critical aspects is presented below.

After the initial study by Defoe [31], RAP was investigated as a component of RAP-VA mixtures. Maher et al. [32] were among the first researchers to evaluate the performance of unbound aggregates containing RAP in terms of Resilient Modulus ($M_R$), which is supposed to better reflect the strength of the unbound materials compared to the Young modulus (E). They concluded that a mixture of VA including RAP resulted in a higher

$M_R$ compared to pure VA. Taha et al. [16] investigated and evaluated the use of RAP as an unbound aggregate in terms of California Bearing Ratio (CBR). It was found that the CBR value decreased as the percentage of RAP was increased from 20% to 100% and none of the samples achieved the desired minimum CBR value, which was 80% in the case of a base course. Therefore, in their conclusions, they suggested that RAP should be used in a proportion lower than 60% in the case of subbase courses, while this proportion was limited to 10% in the case of base courses.

In the 2000s, many researchers continued to evaluate RAP as a base course material and conducted $M_R$ tests. The conclusions of some of these studies [33,34] stated that the $M_R$ of 100% RAP samples is greater than that of a natural aggregate. On the other hand, McGarrah [35] concluded that samples with 100% RAP did not have the same bearing capacity as virgin aggregates, while Kim et al. [23] concluded that a mixture of 50% aggregate and 50% RAP had higher stiffness than samples with 100% aggregate at higher confining pressures. Finally, other researchers [24,36–38] confirmed that RAP materials in a certain ratio were a suitable and sustainable solution for unbound base and subbase courses since they were reported to have equivalent or higher $M_R$ and higher stiffness than a mixture of 100% virgin aggregates. Hoppe et al. [37] emphasized that a proportion of RAP of up to 50% increased the stiffness of the overall base course material, while Nokkaew [38] concluded that the RAP aggregates underwent lower stresses compared to conventional limestone aggregates. A recent study [39] examined RAP-VA blends of four different types of RAP and three different types of VA with respect to $M_R$ and found that a RAP-VA blend had a higher $M_R$ value than VA under the same loading conditions.

Regarding the deformation properties of RAP, some researchers have pointed out that specimens containing RAP appear to develop higher permanent deformation than natural aggregates [23,24,33]. This was confirmed by Dong and Huang [40], who evaluated and compared the properties of RAP, crushed limestone and crushed gravel with similar gradation and degree of compaction. In contrast, Jeon et al. [26] conducted multistage permanent deformation tests of 100% RAP and 100% VA and found that RAP exhibited higher deformation than VA at low stresses, while the opposite was true at higher stresses. Attia [41] also performed triaxial repeated loading tests on three different specimens, namely 100% RAP, 50% RAP −50% VA and 100% VA, and reported that a mixture of RAP and VA exhibited lower permanent deformation than a 100% VA specimen. Arshad and Ahmed [27] came to similar conclusions in that the increase in residual cumulative strains was insignificant when the RAP contents varied from 0 to 50%. Instead, the increase in residual cumulative strains during resilient modulus test was significant at RAP contents of 75% compared to the corresponding values for the fresh granular samples.

In addition, Ullah et al. [25] found that the addition of RAP to VA generally increased permanent deformation and demonstrated other factors affecting the performance of the material. In this case, even with a constant proportion of RAP in the VA mix, the binder content of RAP showed that some of the mixes performed the same or better than VA alone. In another research, Ullah and Tanyu [29] studied the behavior of RAP-VA mixtures when the moisture content was increased from the OMC to OMC + 2% and OMC + 4%, respectively. The results showed that the mixtures with RAP-VA performed better than VA due to their higher potential to excrete excess moisture. Regarding the gradation, Ullah et al. [25] studied the effects of different particle size distributions on the permanent deformations of RAP-VA blends, trying to establish suitable limits for these blends. They found that the addition of RAP to VA reduced the permanent strains to a level comparable to that of 100% VA samples. Pradhan and Biswal [22] also mixed RAP and VA in two different gradations and evaluated their performance based on the CBR value at optimum moisture. Their results showed that the CBR value increased from 32 to 100% when RAP is mixed with VA at a ratio of 45–55%, and they concluded that these RAP-VA blends are suitable for subbase layer. In the same context, it was recently reported that the use of RAP in road construction varies from 10% to 50% for base and subbase layers [42].

Although the incorporation of RAP into the unbound pavement layers has been extensively researched, the results seem to be controversial in the worldwide literature. In particular, regarding the permanent deformations, there are researchers who claim that the developed permanent strains of RAP-VA blends are significant compared to those of pure VA [23–25], while there are studies that contradict the previous results [26–28]. This controversy leads to the fact that there is still room for further research. Therefore, the present study investigates RAP as an unbound material, using a laboratory procedure to determine the physical and mechanical properties of mixtures of RAP and VA, with the aim of determining the optimum proportion of the materials in the blend. The tests used are consistent with standard test procedures used for VA materials.

## 3. Materials and Methods

### 3.1. Materials

For this study, RAP material was taken from the milling process of a pavement section under rehabilitation. Thereafter, RAP was blended with VA in different proportions, while a sample consisting of pure VA was used as a control sample for the comparison of results. Table 1 lists the proportions of RAP-VA in each blend, while the gradation curves of RAP and VA are shown in Figure 1. It should be noted that the proportions in Table 1 were selected based on existing research and/or recommendations [29,42,43] and, considering that the optimization of mix design is desirable, as an excessive amount of one component could disturb the required balance, e.g., desirable physical and mechanical properties [44].

**Table 1.** Proportions of RAP-VA blends.

| Blend | RAP (%) | VA (%) |
| --- | --- | --- |
| A | 0 | 100 |
| B | 10 | 90 |
| C | 20 | 80 |
| D | 30 | 70 |
| E | 40 | 60 |

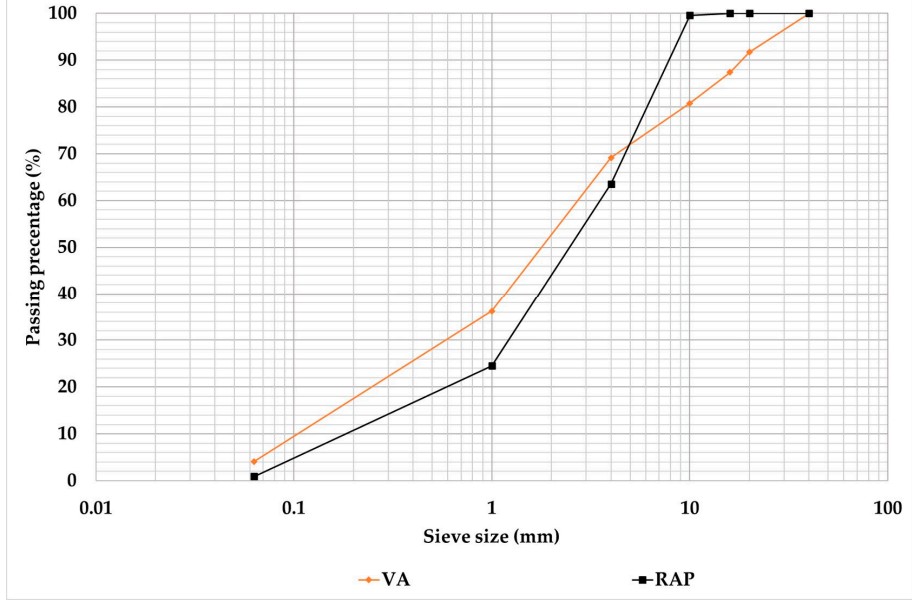

**Figure 1.** Grain size distribution of VA and RAP.

### 3.2. Methodology

#### 3.2.1. General Description

The research methodology included the laboratory-based procedures and the analysis of the experimental results. The experimental process involved the determination of the physical and mechanical properties of the blends under study. In particular, in the case of physical properties, the grain size distribution (i.e., the sieve analysis), the dry density and the moisture content (i.e., the compaction test) were determined. Then, the mechanical properties of the studied blends were determined through CBR tests and triaxial repeated load tests (TRLT). These tests led to the estimation of the CBR and the $M_R$ value, the main parameters used in the design, as well as the permanent deformations. Finally, the laboratory findings were discussed and analyzed to reach the conclusions of this study. In Figure 2, the methodology of the current study is depicted.

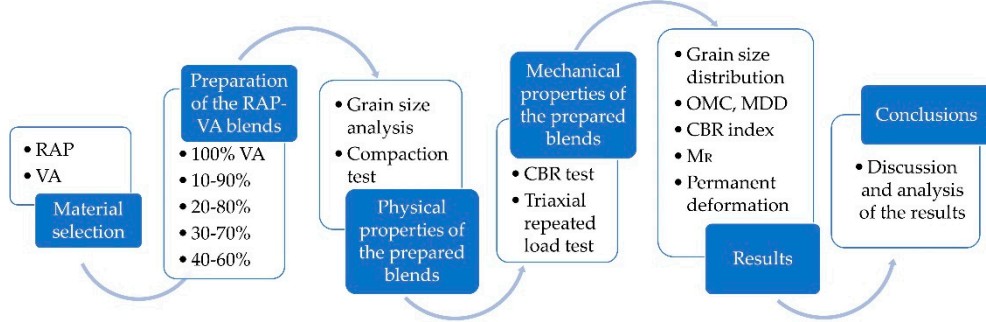

**Figure 2.** Research methodology.

In next sections, the steps of research methodology are presented more thoroughly, while the discussion of the results and the conclusions are following.

#### 3.2.2. Physical Properties: Grain Size Analysis

A sieve analysis was performed according to EN 933-2 [45], so that the gradation of each blend was determined. According to the specifications, the sieves should follow a specific order. In this case, the sieve of 80 mm was not used, as this sieve could not retain any particles of the tested materials. Additionally, for each blend, the coefficient of uniformity $C_u$ and the coefficient of curvature $C_c$ were estimated through Equations (1) and (2), respectively:

$$C_u = \frac{d_{60}}{d_{10}} \tag{1}$$

$$C_c = \frac{d_{30}^2}{d_{10} * d_{60}} \tag{2}$$

where $d_{60}$ is the grain diameter at which 60% of mixture particles are finer; $d_{10}$ is the grain diameter at which 10% of mixture particles are finer; and $d_{30}$ is the grain diameter at which 10% of mixture particles are finer.

These values are used to classify whether the material was well-graded as well as to evaluate the shape of the gradation curves. According to ASTM D2487-17E01 [46], the material was considered to be well-graded if $C_u$ was greater than 4 for coarse material or 6 for finer material, including sand, and $C_c$ was between 1 and 3.

#### 3.2.3. Physical Properties: OMC and Dry Density

A compaction test was performed to determine the Optimum Moisture Content (OMC) of each studied blend. The test was based on the modified Proctor test of the European specification EN 13286-2 [47]. For this test, 6 kg of each studied blend was used. The material was placed in the mold in three layers. For each layer, a mass of about 1.6 to 1.8 kg was used, while 28 blows were applied with the rammer of 4.5 kg to compact

each layer and then determine the dry density. This procedure was repeated for different values of moisture content to determine the OMC as the moisture content corresponding to the MDD.

### 3.2.4. Mechanical Properties: CBR

CBR tests were performed in accordance with EN 13286-47 [48]. Once the OMC was determined for each blend, six new specimens for each RAP-VA composition were again compacted at the OMC, so that the CBR test could follow. Each specimen was placed on the loading device. During the test, the load was applied to the specimen through the penetration piston, so that the penetration rate was constant at about 1.27 mm/min. The load value was recorded for each penetration into the material. The force/penetration curve was recorded for each test.

The *CBR* value for each desired penetration is estimated using Equation (3):

$$CBR = \frac{P}{P_T} \times 100\% \tag{3}$$

where $P$ is the applied load which causes penetration to the sample equal to 2.5 mm and 5 mm, respectively, and $P_T$ is the standard load (13.2 and 20 kN) corresponding to penetration of 2.5 and 5 mm, respectively.

### 3.2.5. Mechanical Properties: Resilient Modulus and Permanent Deformation

In this study, the evaluation of mechanical behaviors of the unbound materials under investigation was completed in accordance with AASHTO T307-99 [49]. According to [49], all base and subbase materials were classified as Material Type 1 and Material Type 2. In this study, the investigated blends met the requirements of Material Type 1 because the proportion of material passing through the 2-mm sieve was less than 70% and the proportion of material passing through the 0.075-mm sieve was less than 20%. As for sample preparation, a moisture content equal to OMC was considered for all blends, since this was the target moisture content in the field for unbound pavement layers, according to the practice for conventional materials [22]. After the assembly at the bottom of the mold, the material with OMC was compacted in 5 layers, each 6 cm thick, using a vibratory compaction device. Finally, the specimen was assembled at the top, installed in the triaxial chamber and the test started. As for the loading conditions, the specimens were imposed to an initial load of 103.4 kPa (confining pressure and maximum axial stress), while the cycle stress was set to be 93.1 kPa, in order to eliminate the effects of the interval between compaction and loading and between initial loading and reloading. Afterwards, the value of confining pressure was set to be 20.7 kPa, which was increased in every three sequences.

The applied stress and the developed strains describe the mechanical behavior of the material [50], as it is shown in Figure 3.

For the estimation of the resilient modulus $M_R$, the experimental data of the last five loading cycles were taken into account. The $M_R$ value for each loading cycle was estimated through Equation (4):

$$M_R = \frac{S_{cyclic}}{e_r} \tag{4}$$

where $S_{cyclic}$ is applied cyclic stress; $e_r$ is the elastic deformation; and $S_{cyclic}$ is estimated through Equation (5):

$$S_{cyclic} = \frac{P_{cyclic}}{A} \tag{5}$$

where $P_{cyclic} = P_{max} - P_{contact}$ ($P_{max}$ is the maximum applied axial force and $P_{contact}$ is the contact load, which maintains a positive contact between the specimen cap and the specimen) and $A$ is the initial cross-sectional area of the specimen.

The permanent deformations were determined directly from the experimental results by accumulation. The concept of accumulation of permanent deformations with increasing number of loading cycles is shown in Figure 3.

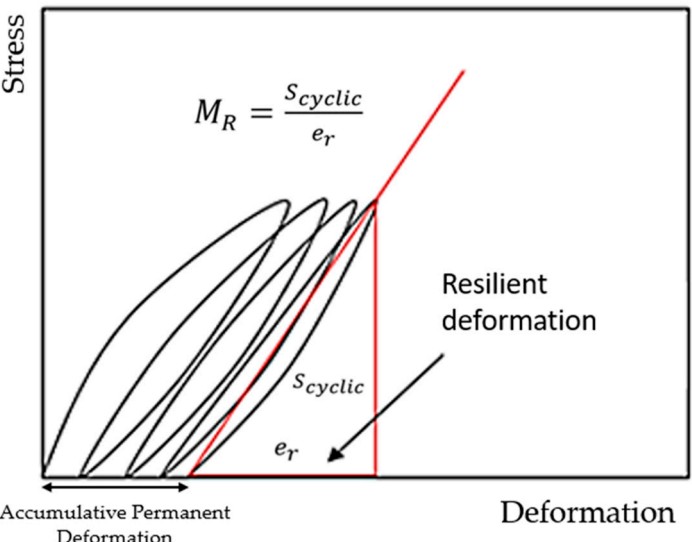

**Figure 3.** Mechanical behavior of granular material under triaxial testing [50].

## 4. Results and Discussion

### 4.1. Physical Properties

Following the grain size analysis for all tested blends, the grain size distribution for each of them is presented in Figure 4.

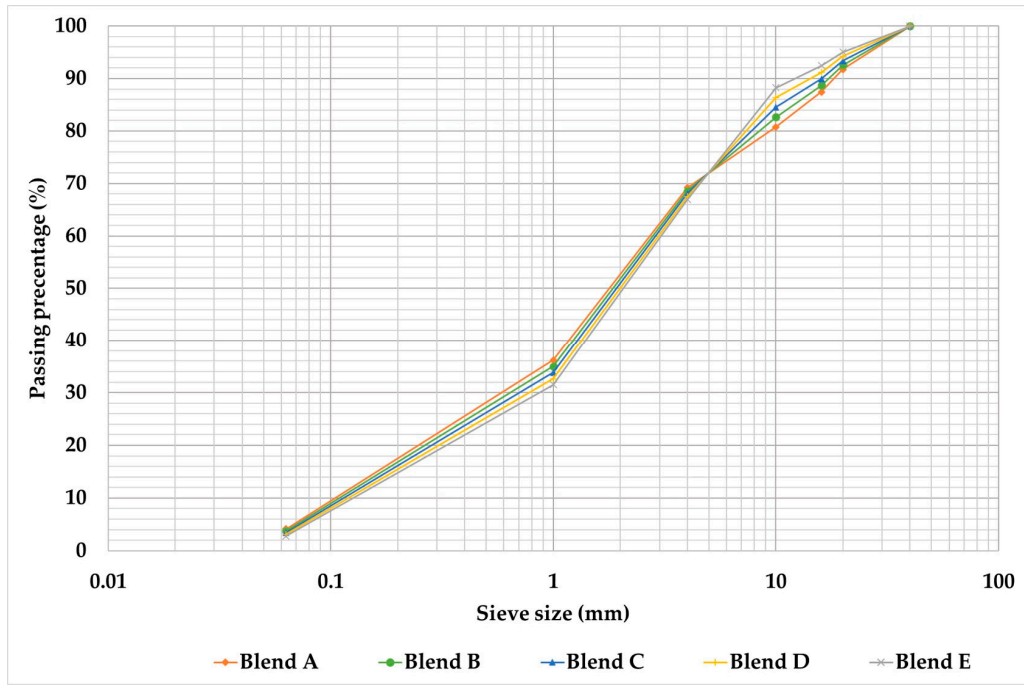

**Figure 4.** Grain size distribution of all examined blends.

As can be seen from Figure 4, all samples have a similar grain size distribution with slight differences. So, it is obvious that the RAP-VA blends (B–E) are comparable to VA (blend A) and that it is reasonable to proceed to the next steps of laboratory investigation. In addition, all the materials are considered to be well graded, since the coefficients $C_u$ and $C_c$

are within the specified limits. The values of Cu and Cc are shown in Table 2. In particular, the value of $C_u$ decreases with the increasing content of RAP, but is still well above the specified limit. The value of $C_c$ generally increases—within the limits—as the content of RAP increases. This means that the addition of RAP can also improve the gradation of VA, because the closer the Cc value is to the specified limits, the flatter the gradation curve becomes. Moreover, the values of the coefficients for the blends RAP-VA are close to those of the blend VA, which means that the addition of RAP does not significantly change the grain size distribution of the control blend (100% VA).

**Table 2.** Coefficients $C_U$ and $C_C$.

| Coefficients | Blend A | Blend B | Blend C | Blend D | Blend E |
|:---:|:---:|:---:|:---:|:---:|:---:|
| $C_U$ * | 29 | 40 | 30 | 23 | 23 |
| $C_C$ ** | 1.24 | 1.225 | 1.87 | 1.64 | 2.07 |

* $C_U$: Coefficient of uniformity, ** $C_C$: Coefficient of curvature.

As for the content of RAP, RAP contains mainly finer particles, as the material passes through sieves with openings of 10 mm and below. Moreover, it can be observed in the studied blends that when the sieve size exceeds 4 mm, the percentage of grains passed increases with the content of RAP. In contrast, when the sieve size is less than 4 mm, the percentage of grains passed decreases with the increasing content of RAP. In the first case, the grain size distribution of blend A (100% VA) is located at a lower position, while in the second case it is located at a higher position. It could be concluded that, as the percentage of RAP increases, the blend consists of finer particles, which could be due to the extraction process of the material, i.e., milling and grinding. This confirms that RAP is a fine material.

Based on the compaction test, the values of dry density versus moisture content are estimated. As mentioned earlier, OMC corresponds to MDD. Detailed results for all blends can be found in Table 3. Since all OMC values were around 6% without varying significantly (an average OMC value of 6.3% with a coefficient of variation of 6.5%), a fixed OMC value of 6% was considered representative of all blends for the remainder of the laboratory study. Therefore, this value was used to repeat the compaction procedure prior to the CBR tests.

**Table 3.** Results from the modified Proctor test for all blends.

| | Blend A | Blend B | Blend C | Blend D | Blend E |
|:---:|:---:|:---:|:---:|:---:|:---:|
| OMC * | 6.9% | 6.2% | 6.0% | 6.5% | 5.9% |
| MDD ** (kg/m$^3$) | 2196 | 2130 | 2083 | 2065 | 2041 |

* OMC: Optimum Moisture Content, ** MDD: Maximum Dry Density.

It is worth noting that the OMC values derived from the tests are considered typical. For example, in the worldwide literature [22,25,28,29], it is stated that the moisture content of the RAP-VA blends varies between 4.5 and 9%, while an OMC value of 5.5 to 7% is commonly used. Accordingly, MDD values are also considered typical. Many researchers [22,25,28–30] found that the MDD for RAP-VA blends ranges from 2050 to 2350 kg/m$^3$. Overall, it appears that the addition of RAP to VA does not significantly alter the physical properties of VA for use in unbound pavement layers. The gradation coefficients are within the limits and the OMC and MDD parameters have similar values for the VA and RAP-VA blends.

### 4.2. CBR Results

According to EN 13286-47 [48], a correction to the force–penetration curve may be required before calculating the CBR value for each blend. If the initial portion of the curve is concave upward, which may be due to an irregular surface, a correction is made by drawing a tangent to the curve at the point of greatest slope. In this way, the measurement

of penetration begins at the intersection of this tangent with the penetration axis. Finally, the CBR value for a penetration depth of 2.5 mm and 5 mm is calculated using Equation (3), and the final value is the highest of these values. The final CBR values (average values) for each mixture are shown in Figure 5.

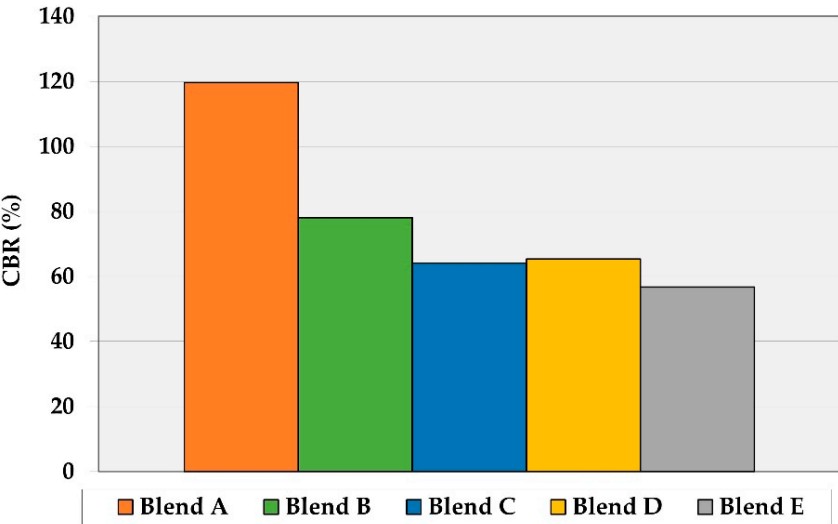

**Figure 5.** CBR values for each blend.

As can be seen in Figure 5, the CBR value of blend A (VA) is 120%, which seems irrational because it is above the upper limit of 100%. However, if the material tested is crushed limestone that is well compacted, CBR values above 100% are not uncommon [51]. Such a material is blend A. In addition, this value may result from the presence of coarser particles under the penetration piston during the test. Furthermore, the addition of 10% RAP to VA leads to a reduction in the CBR value by up to 35%, starting from a CBR value of 120% in the case of pure VA (blend A). This reduction reaches up to 50% when 20% RAP is added to VA (blend B). Beyond that, the addition of RAP does not lead to further reductions, since the CBR values of blends C, D and E are almost the same, ranging from 57 to 65%. The slight differences between blends C, D and E could be explained by the better interlocking of the particles of blend D and/or the coarser material that was under the penetration piston during the CBR test, which depends on the sample preparation. In addition, the decrease in the CBR value due to the addition of RAP to VA can be explained by the nature of the CBR test. It is based on a penetration procedure in a limited area as large as the contact area between the piston and the sample. Therefore, a finer material such as blend B, and especially blend D and E, is not able to resist penetration due to the limited interlocking of the individual particles. In this context, Figure 5 shows that the CBR value generally decreases with the increasing content of RAP.

Finally, it is worth mentioning that the observed reduction in the CBR value by adding RAP to VA confirms the scientific findings in the international literature related to the presence of RAP in unbound pavement layers [16,23,35].

### 4.3. Triaxial Test Results

#### 4.3.1. Resilient Modulus

The estimation of $M_R$ values is based on the AASHTO T307-99 [49], also taking into account the international literature [52,53]. Table 4 shows the final values of $M_R$, which correspond to the maximum axial loads (Smax) of all blends for each sequence of 100 cycles. Overall, the values of $M_R$ for each blend increase as the force is applied during the load cycles in each sequence. It is noteworthy that in blend E the highest value of $M_R$ occurs in the penultimate sequence. This could be due to the fact that blend E contains finer particles compared to the other blends and may need to be subjected to additional or higher

loading to achieve the highest degree of compaction between particles. The influence of the compaction process on the mechanical behavior has also been mentioned elsewhere [5].

**Table 4.** Results of the TRL tests regarding the $M_R$ calculation for all blends.

| Sequence No. | A | | B | | C | | D | | E | |
|---|---|---|---|---|---|---|---|---|---|---|
| | $M_R$ (MPa) | $S_{max}$ (kN) | $M_R$ (MPa) | $S_{max}$ (kN) | $M_R$ (MPa) | $S_{max}$ (kN) | $M_R$ (MPa) | $S_{max}$ (kN) | $M_R$ (MPa) | $S_{max}$ (kN) |
| 1 | 86.88 | 20.91 | 66.46 | 20.93 | 111.65 | 20.82 | 119.85 | 20.82 | 129.78 | 20.96 |
| 2 | 80.85 | 41.14 | 88.36 | 41.14 | 127.60 | 41.14 | 127.17 | 41.15 | 148.19 | 41.26 |
| 3 | 84.00 | 61.41 | 98.47 | 61.05 | 137.55 | 61.52 | 137.76 | 61.58 | 158.46 | 61.47 |
| 4 | 94.52 | 34.86 | 81.13 | 34.93 | 150.09 | 34.89 | 157.07 | 34.90 | 192.03 | 34.89 |
| 5 | 103.83 | 68.29 | 100.50 | 68.03 | 178.17 | 68.12 | 177.42 | 68.11 | 206.17 | 68.01 |
| 6 | 120.39 | 101.53 | 109.25 | 101.47 | 191.95 | 101.65 | 182.20 | 101.67 | 212.84 | 101.37 |
| 7 | 217.20 | 69.70 | 109.26 | 69.73 | 284.59 | 69.71 | 262.52 | 69.66 | 287.88 | 69.71 |
| 8 | 258.12 | 134.41 | 153.08 | 134.21 | 322.68 | 134.40 | 290.74 | 134.33 | 327.16 | 134.42 |
| 9 | 293.16 | 199.00 | 189.08 | 199.12 | 339.26 | 198.84 | 316.06 | 199.15 | 342.86 | 199.03 |
| 10 | 271.82 | 69.91 | 118.77 | 69.89 | 343.25 | 69.92 | 340.80 | 70.06 | 348.23 | 69.85 |
| 11 | 287.07 | 101.56 | 136.83 | 101.51 | 364.08 | 101.76 | 363.45 | 101.75 | 372.16 | 101.57 |
| 12 | 341.26 | 199.02 | 213.13 | 198.73 | 406.55 | 198.82 | 393.75 | 198.92 | 406.59 | 198.74 |
| 13 | 340.84 | 104.22 | 158.68 | 104.07 | 420.92 | 103.94 | 420.30 | 103.97 | 425.91 | 104.26 |
| 14 | 351.91 | 135.01 | 175.66 | 134.84 | 436.23 | 134.92 | 436.50 | 134.94 | 442.02 | 134.92 |
| 15 | 406.99 | 261.75 | 252.26 | 262.02 | 464.49 | 262.10 | 454.35 | 262.17 | 415.29 | 262.15 |

Figure 6 shows the $M_R$–$S_{max}$ curves for all blends. For each blend, the trend line was selected based on the highest and best fitted coefficient of determination $R^2$ [54,55]. All follow a polyonymic equation, while only the equation for blend B is nearly linear. Blends A, C, D and E have a similar $R^2$ around 0.70, while blend B (10–90% RAP-VA) has an $R^2$ of 0.94. This—combined with the linearity of the equation for blend B—may not be a realistic condition for unbound materials, as they exhibit nonlinear behavior [50]. Moreover, the $M_R$ values of blend B are the lowest among the other blends, which was not expected since its CBR value is the highest. Accordingly, the addition of only 10% RAP to VA cannot explain such a reduction in the bearing capacity to this extent, since the other blends with a higher RAP content exhibited even higher $M_R$ values in each sequence than the VA blend. Therefore, it was decided to exclude blend B from the additional investigation in terms of the permanent deformation assessment.

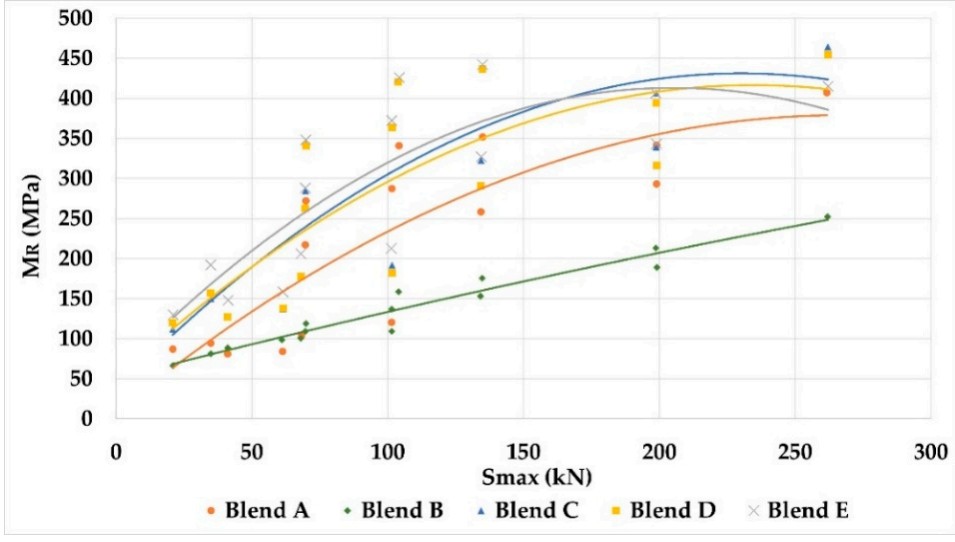

**Figure 6.** $M_R$ values regarding the maximum applied axial stress $S_{max}$.

In addition, Figure 6 shows a vertical alignment of the measurements. This observation is due to the applied stress in each sequence, since the process follows a sequence of loading and unloading conditions. Therefore, some values of applied stress are almost the same. A similar up and down is observed in the values of $M_R$, but the range has a lower value, with the exception of blend B. Moreover, the increasing proportion of RAP in the RAP-VA blends leads to an increase in the $M_R$ value in each sequence. This could be due to the presence of asphalt which bonds the aggregates and the fact that the voids are filled with the finer particles of RAP. Finally, the observed results for the $M_R$ values seem to be reasonable and are in agreement with previous studies [19,27,39] in which it was observed that the $M_R$ parameters of the RAP-VA blends mainly have a value of 400 MPa and above. Possible differences could mainly depend on the quality of RAP material used in RAP-VA blends.

Overall, all RAP-VA blends have a $M_R$ value greater than the pure VA value, given the last sequence and the end of the $M_R$ test. More specifically, the 20–80% RAP-VA blend (blend C) has the highest $M_R$ value of all the others, corresponding to 464 MPa. The 30–70% RAP-VA blend (blend D) follows with a slightly lower $M_R$ value of 454 MPa and the last one is the 40–60% RAP-VA blend (blend E) with a $M_R$ value of 415 MPa. This observation proves the influence of the RAP content on the $M_R$ modelling. Thus, it can be seen that the RAP-VA blends perform satisfactorily in terms of stiffness and bearing capacity. This is significant considering that the stiffness of the unbound layers is an important property to consider in both pavement design and analysis, with the usual range of values for unbound layers being between 250 and 600 MPa.

### 4.3.2. Permanent Deformation

From the triaxial test, the developed permanent deformations are determined for each blend. Figure 7 shows the accumulated permanent strains for the blends A, C, D and E as a function of loading cycles.

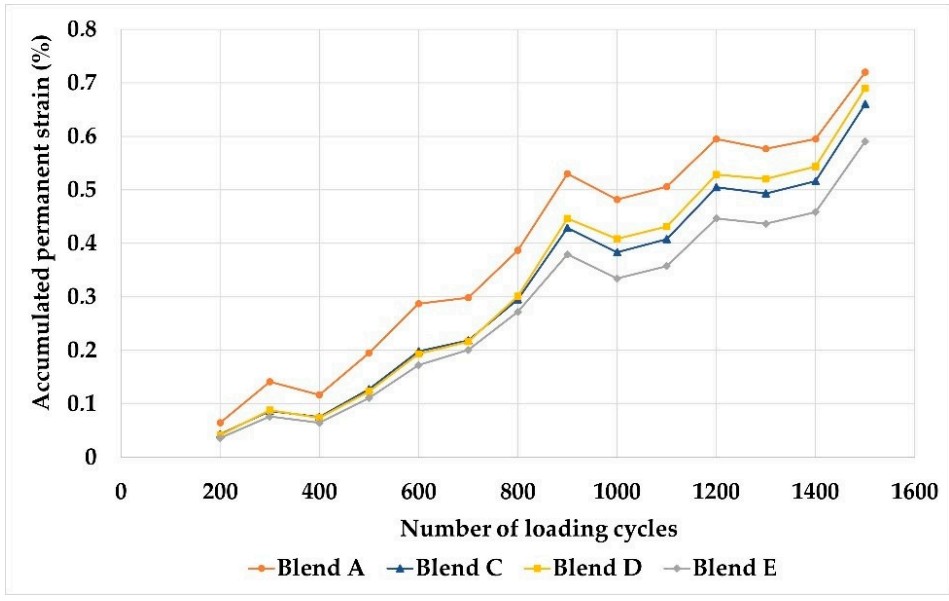

**Figure 7.** Permanent strains regarding the loading cycles for blends A, C, D and E.

It can be seen that the permanent strains for all blends increase as the number of loading cycles increases, with all specimens showing similar trends in their curves every 100 cycles. The accumulated permanent strain after 1500 loading cycles is 0.72, 0.66, 0.69 and 0.59% for blend A, C, D and E, respectively. Therefore, blends with a percentage of RAP develop lower permanent deformations than the blend with 100% VA (blend A). Moreover, blend E (40–60% RAP-VA) develops the lowest permanent strains among all blends, as well as among all blends with a proportion of RAP. In addition, blend D (30–70% RAP-VA) develops a higher permanent strain than blend C (20–80% RAP-VA), which consists of a

lower RAP content, an observation that does not allow specific conclusions to be drawn about the behavior of RAP-VA blends as unbound materials. Nevertheless, it can be said that the addition of RAP is acceptable for use in unbound pavement layers, since RAP particles have been already used in pavement layers, so they have been already exposed to some extent to the cumulative permanent deformation that results in a lower rate of the additional cumulative permanent deformation. Overall, the performance of unbound RAP blends is an area that should be investigated, as some controversial results can be found in the international literature. For example, higher permanent deformations have been reported for RAP-VA compared to VA [23–25], while the opposite has been reported elsewhere [26–28]. The latter is also confirmed by the current experimental results. Thus, it seems that this is an aspect that needs further investigation.

Overall, according to the current experimental procedure, it is concluded that a RAP-VA blend containing up to 40% RAP can be incorporated into the base course and subbase layer, since its physical properties are similar to those of pure VA and its mechanical properties appear to be significantly better than those of conventional aggregates. In particular, the corresponding $M_R$ values are in the range of 410–470 MPa, which seems acceptable considering that the $M_R$ value of the tested pure VA is 406 MPa and the usual range of values for use in unbound layers is between 250 and 600 MPa. As far as permanent deformations are concerned, all tested RAP-VA blends show lower cumulative permanent deformation than pure VA. As for the CBR value, the last observation seems to contradict the decrease in the CBR value with increasing RAP content, but the two parameters are determined by tests of a different nature; the CBR test is a single point penetration, while the triaxial test is based on lateral and axial compression by a variable stress.

## 5. Conclusions and Future Recommendations

This study focused on investigating the performance of RAP for use in unbound pavement layers. To this end, an experimental procedure was used to determine the physical and mechanical properties of blends of RAP and virgin aggregates (VA). Laboratory tests included grain size analysis, a modified Proctor compaction test, a CBR test and triaxial repeated load tests to determine $M_R$ and deformation properties. This procedure was applied to five different blends: A (pure VA), B (10–90% RAP-VA), C (20–80% RAP-VA), D (30–70% RAP-VA) and E (40–60%). Finally, the following conclusions are drawn:

. Compared to pure VA, the addition of RAP to VA results in a somewhat finer blend, while the gradation and compaction properties of RAP-VA blends are acceptable and within the specified limits;
. The addition of RAP to the blend lowers the CBR value and this reduction reaches up to 50% when 40% RAP is added to VA;
. The value of $M_R$ is higher than that of pure VA in all the RAP-VA blends studied;
. All the blends with RAP develop lower permanent deformations than the blend with 100% VA for the same loading history of 1500 cycles. In particular, blend E with the higher percentage of RAP has the lowest deformation value among all the blends;
. Based on the current experimental findings, it seems that selecting RAP proportions up to 40% is a rational approach to develop RAP-VA mixes that are to be incorporated into unbound pavement layers, since the physical and especially mechanical properties of RAP-VA blends are similar or better than natural aggregates.

Considering all the controversies in the existing literature on the performance of RAP-VA blends, the results of the present study provide guidance on practical issues in the use of RAP materials for pavement construction. However, they are only recommendatory in nature, which means that testing is required each time to define the capabilities of RAP, taking into account local effects and material conditions. This, in turn, could lead to the desired environmental outcomes: reduced energy consumption and GHG emissions, reduced need for RAP stockpiling and reduced construction and rehabilitation costs.

In practice, the equivalent load cycles over the life of a pavement are much higher, so further investigation is needed to determine the development of permanent deformations

of RAP-VA blends in terms of pavement sustainability. The development of predictive models that take into account various inherent properties of materials, the content of RAP and various simulations of environmental and traffic conditions in the field is another area for future research following other similar studies. It may also be useful to conduct several cycles of a full experimental program for RAP-VA blends with different proportions of RAP to provide a statistical perspective on the behavior of RAP in unbound layers. In addition, further laboratory testing of RAP-VA blends with RAP proportions of 50% or more should be conducted, focusing on a more comprehensive understanding of bearing capacity as expressed by CBR and $M_R$ tests (taking into account additional aspects such as the effect of RAP particle fragmentation, stress conditions, etc.) and deformation characteristics. This could lead to a more comprehensive and systematic use of RAP in road construction. Finally, it is equally important that future research focus on life-cycle cost analysis to demonstrate the cost effectiveness of these alternative materials, namely RAP, compared to conventional materials.

**Author Contributions:** Conceptualization, C.P.; methodology, C.P.; testing and analysis, M.T. and C.P.; writing—original draft preparation, M.T. and K.G.; writing—review and editing, C.P., K.G. and M.T. All authors have read and agreed to the published version of the manuscript.

**Funding:** This research received no external funding.

**Institutional Review Board Statement:** Not applicable.

**Informed Consent Statement:** Not applicable.

**Data Availability Statement:** Not applicable.

**Conflicts of Interest:** The authors declare no conflict of interest.

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
