# Peer review of "Physical and Mechanical Properties of Reclaimed Asphalt Pavement (RAP) Incorporated into Unbound Pavement Layers"

_applsci, doi:10.3390/app13010362_

Round 1
Reviewer 1 Report
The paper investigated the feasibility of recycled asphalt pavement as unbound granular material by laboratory experiments. It is an interesting topic in pavement engineering. However, from a scientific point of view, it will have to undergo indispensable major revisions before it can be considered for publication. Here are some comments for the authors' consideration.
1. The review on engineering properties of C&D materials is insufficient. In fact, there are many literatures on these topics. It is necessary for authors to review and cite them appropriately. For instance:
Zhang J, Zhang A, Li J, et al. Gray correlation analysis and prediction on permanent deformation of subgrade filled with construction and demolition materials[J]. Materials, 2019, 12(18): 3035.
Yao Y, Li J, Liang C, et al. Effect of coarse recycled aggregate on failure strength for asphalt mixture using experimental and dem method[J]. Coatings, 2021, 11(10): 1234.
2. In the section 3.1, please display the physical and mechanical properties of RAP and VA
3. In the section 3, please delete the figures 1(b) and 2(b).
4. In the section 3.5, the mechanical behaviors of unbound base layer have significant dependence on the moisture and stress conditions. Why did the author only consider the OMC? Meanwhile, some typical models may be useful for these triaxial testing analysis, e.g.,
Li J, Zhang J, Zhang A, et al. Evaluation on deformation behavior of granular base material during repeated load triaxial testing by discrete-element method[J]. International Journal of Geomechanics, 2022, 22(11): 04022210.
Asefzadeh A, Hashemian L, Bayat A. Characterization of permanent deformation behavior of silty sand subgrade soil under repeated load triaxial tests[J]. Transportation Research Record, 2017, 2641(1): 103-110.
5. In section 4.1, please add all the Proctor curves from Blend A to Blend E.
6. In section 4.2, the effect of particle fragmentation on CBR values should be considered. It is recommended to compare the gradation before and after CBR.
7. In section 4.3, the estimation models should be applied to the analysis of the results. How is the effect of RAP content considered in the model?
Author Response
The authors would like to thank the reviewer for his/her valuable comments. Detailed answers are provided below.
The paper investigated the feasibility of recycled asphalt pavement as unbound granular material by laboratory experiments. It is an interesting topic in pavement engineering. However, from a scientific point of view, it will have to undergo indispensable major revisions before it can be considered for publication. Here are some comments for the authors' consideration.
- The review on engineering properties of C&D materials is insufficient. In fact, there are many literatures on these topics. It is necessary for authors to review and cite them appropriately. For instance:
- Zhang J, Zhang A, Li J, et al. Gray correlation analysis and prediction on permanent deformation of subgrade filled with construction and demolition materials[J]. Materials, 2019, 12(18): 3035.
- Yao Y, Li J, Liang C, et al. Effect of coarse recycled aggregate on failure strength for asphalt mixture using experimental and dem method[J]. Coatings, 2021, 11(10): 1234.
Reply: Thank you for the comment. Please see lines 36-39 and references 1 and 3.
- In the section 3.1, please display the physical and mechanical properties of RAP and VA
Reply: Thank you for the comment. Please see lines 152-157 and Figure 1, 8 -10.
- In the section 3, please delete the figures 1(b) and 2(b).
Reply: Thank you for the comment. Figures 1(b) and 2(b) were removed.
- In the section 3.5, the mechanical behaviors of unbound base layer have significant dependence on the moisture and stress conditions. Why did the author only consider the OMC?
Reply: Thank you for the comment. Please see lines 225-227 and 230-235.
Meanwhile, some typical models may be useful for these triaxial testing analysis, e.g.,
- Li J, Zhang J, Zhang A, et al. Evaluation on deformation behavior of granular base material during repeated load triaxial testing by discrete-element method[J]. International Journal of Geomechanics, 2022, 22(11): 04022210.
- Asefzadeh A, Hashemian L, Bayat A. Characterization of permanent deformation behavior of silty sand subgrade soil under repeated load triaxial tests[J]. Transportation Research Record, 2017, 2641(1): 103-110.
Reply: Thank you for the comment. Please see lines 341-342 and references 47 and 48.
- In section 4.1, please add all the Proctor curves from Blend A to Blend E.
Reply: Thank you for the comment. The Figure with proctor curves has been removed and the requested information is shown in Table 3.
- In section 4.2, the effect of particle fragmentation on CBR values should be considered. It is recommended to compare the gradation before and after CBR.
Reply: Thank you for the comment. Please see lines 461-466.
- In section 4.3, the estimation models should be applied to the analysis of the results. How is the effect of RAP content considered in the model?
Reply: Thank you for the comment. Please see lines 383-384 and 456-459.
Reviewer 2 Report
This manuscript is unable to provide any novel results and it appears to be a simple testing report rather than a research paper. Apart from the results, no scientific discussion or advancement of the subject matter is being rendered; thus, this article is of no interest to the scientific community.
Author Response
The authors would like to thank the reviewer for his/her valuable comments. Detailed answers are provided below.
This manuscript is unable to provide any novel results and it appears to be a simple testing report rather than a research paper. Apart from the results, no scientific discussion or advancement of the subject matter is being rendered; thus, this article is of no interest to the scientific community.
Reply: Thank you for the comments. The authors have incorporated several amendments according to all reviewers’ comments. Please see the colored parts in the revised version of the manuscript.
Reviewer 3 Report
The manuscript deals with recycled asphalt pavement (RAP) as an unbound pavement material. The research is not innovative but it is useful. The experimental program is limited. The structure of the manuscript can be improved. The results are well explained and discussed. Some of the important points are highlighted below:
Comment 1: The abstract should state the purpose of the research, the principal results, and the major conclusions briefly. An abstract is often presented separately from the article, so it must be able to stand alone. The abstract structure should be as follows: (introduction, problem of statement, materials, methods, results, and recommendations). Please revise your abstract.
Comment 2: I suggest the authors to conduct a more in-depth review and summarize in section 2.
Comment 3: I suggest the authors to add a citation in Fig. 4.
Comment 4: I suggest the authors to correct the references style.
Comment 5: I suggest the authors to modify section 3 name to materials and methods then 3.1 materials then 3.2 methods (optional) then 3.3 experimentals then 3.3.1 test1, 3.3.2 test2 …
Comment 6: The points presented in the conclusion section are not up to the mark. The authors are advised to revise it completely and try to present information, which is a comprehensive summary of the important aspects discussed in the preceding sections. It certainly lacks in its current form.
· Comparison with previous studies must mention in the result section, not in the conclusion section.
· Delete the citations in the conclusion.
· Mention your own results only.
· Mention recommendations.
· Make the conclusion in points with short details.
· Change the name of the section to “Conclusions and Recommendations”.
Author Response
The authors would like to thank the reviewer for his/her valuable comments. Detailed answers are provided below.
The manuscript deals with recycled asphalt pavement (RAP) as an unbound pavement material. The research is not innovative, but it is useful. The experimental program is limited. The structure of the manuscript can be improved. The results are well explained and discussed. Some of the important points are highlighted below:
- Comment 1: The abstract should state the purpose of the research, the principal results, and the major conclusions briefly. An abstract is often presented separately from the article, so it must be able to stand alone. The abstract structure should be as follows: (introduction, problem of statement, materials, methods, results, and recommendations). Please revise your abstract.
Reply: Thank you for the comment. Please see lines 8-23.
- Comment 2: I suggest the authors to conduct a more in-depth review and summarize in section 2.
Reply: Thank you for the comment. Please see lines 104-106, 116-121, 129-149 and reference list.
- Comment 3: I suggest the authors to add a citation in Fig. 4.
Reply: Thank you for the comment. Please see Figure 5 and line 239.
- Comment 4: I suggest the authors to correct the references style.
Reply: Thank you for the comment. References style was fixed. Please see lines 480-628.
- Comment 5: I suggest the authors to modify section 3 name to materials and methods then 3.1 materials then 3.2 methods (optional) then 3.3 experimentals then 3.3.1 test1, 3.3.2 test2 …
Reply: Thank you for the comment. Please see lines 150, 151, 163, 171, 188, 203 and 218.
- Comment 6: The points presented in the conclusion section are not up to the mark. The authors are advised to revise it completely and try to present information, which is a comprehensive summary of the important aspects discussed in the preceding sections. It certainly lacks in its current form.
- Comparison with previous studies must mention in the result section, not in the conclusion section.
Reply: Ok, it’s done. Please see lines 293-297, 335-337, 372-376, 407-412 and the revised section 5 “Conclusions and Future Recommendations”.
- Delete the citations in the conclusion.
Reply: Ok, it’s done. Please see the revised section 5 “Conclusions and Future Recommendations”.
- Mention your own results only.
Reply: Ok, it’s done. Please see lines 433-447.
- Mention recommendations.
Reply: Ok, it’s done. Please see lines 443-447.
- Make the conclusion in points with short details.
Reply: Ok, it’s done. Please see lines 433-447.
- Change the name of the section to “Conclusions and Recommendations”.
Reply: Ok, it’s done. Please see line 425.
Reviewer 4 Report
The manuscript discusses a laboratory study to evaluation of RAP. It is believed by the authors that the material is sustainable for construction of pavement layers. The paper needs some improvement before it can be considered for publication. Consider my comments.

Author Response
The authors would like to thank the reviewer for his/her valuable comments. Detailed answers are provided below.
- Abstract: add key specific results in abstract
Reply: Thank you for the comment. Please see lines 8-23.
- Abstract: Briefly discuss the motivation – a one line would be good
Reply: Thank you for the comment. Please see lines 8-11.
- Merge introduction and background of this work
Reply: Thank you for the comment. The title of section 2 “Background” has been replaced by “Review of related studies”.
- “Reclaimed Asphalt Pavement (RAP) is one of the waste materials…”: how much quantity is available and how much is produced annually?
Reply: Thank you for the comment. Please see lines 57-58 and reference 11.
- “…which was not observed in the 107 other two cases (OMC+2%, OMC+4%).”: Why 4%?
Reply: Thank you for the comment. Please see reference 32.
- “Proportions of RAP-VA blends.”: Why these blending ratios?
Reply: Thank you for the comment. Please see lines 137-139 and 156-157.
- “Then, the mechanical properties of the studied blends are determined by CBR tests and triaxial repeated load 119 tests (TRLT).”: Why these properties are determined, is this standard procedure for characterization?
Reply: Thank you for the comment. Please see lines 148-149.
- Coefficients Cu and Cc: What is the significance of these coefficients?
Reply: Thank you for the comment. Please see lines 263-270.
- “A compaction test was performed to determine the Optimum Moisture Content 140 (OMC) and the Maximum Dry Density (MDD) of each blend studied.”: What is the value of OMC?
Reply: Thank you for the comment. Please see lines 285-289 and Table 3.
- Figure 1(a) is low quality. Improve to more than 300 dpi.
Reply: Thank you for the comment. The Figure is fixed. Please see line 193 and Figure 2.
- “Where P is the applied load which causes penetration to the sample equal to 2.5 mm and 5 mm respectively, and PT is the standard load (13.2 and 20 kN) corresponding to penetration of 2.5 and 5 mm respectively.”: How much (penetration) is allowed?
Reply: Thank you for the comment. Please see reference 40.
- “Results and discussion”: Discuss and relate all your results with existing literature
Reply: Thank you for the comment. Please see lines 265-270, 275-283, 285-289, 293-297, 317-321, 326-337, 345-350, 365-376 and 383-388.
- About grain size distribution of the blends: All blends are almost the same.
Reply: Thank you for the comment. Please see lines 262-263.
- “Table 2: Coefficients Cu and Cc”: which one is better?
Reply: Thank you for the comment. Please see lines 263-270 and Table 2.
- Table 2: formatting error
Reply: Thank you for the comment. The error is fixed. Please see Table 2.
- “Figure 6 Proctor curve for blend A (100% VA) and blend C (20-80% RAP-VA)”: write sub captions a) and b) and also in Figure as well
Reply: Thank you for the comment. The Figure with proctor curves has been removed and the requested information is shown in Table 3.
- About CBR results for blend A: How 120% is achieved?
Reply: Thank you for the comment. Please see lines 317-321 and reference 44.
- “Figure 9 MR values regarding the maximum applied axial stress Smax”: what is the significance, many points are aligning vertically. Explain in detail
Reply: Thank you for the comment. Please see lines 365-369.
- Did the deformation stop after 1500 cycles?
Reply: Thank you for the comment. Please see lines 380.
- Add limitations of this work, future work and recommendations
Reply: Thank you for the comment. Please see lines 454-469.
- “In summary, …”: Merge this with above conclusions. There is no need to add summary separately.
Reply: Thank you for the comment. Please see lines 448-469.
- References: Check formatting and information of all papers cited. Add review of latest publications.
Reply: Thank you for the comment. Please see the revised reference list.
Reviewer 5 Report
1. The title is too general and common. The author should make it focus or relate with the testing used
2. The Abstract should include General information, main objective, significant methodology, significant results and conclusion
3. The manuscript needs extensive revision for language and grammar especially in experimental sections.
4. The innovations of this manuscript are limited. Most of the results have already been described in some article with the same topics
5. the results is not discussed deeply
6. More discussion and analysis should be included in this article to be published in this journal. experimental section also should be improved and explain in detail.

Author Response
The authors would like to thank the reviewer for his/her valuable comments. Detailed answers are provided below.
- The title is too general and common. The author should make it focus or relate with the testing used
Reply: Thank you for the comment. Please see the revised title.
- The Abstract should include General information, main objective, significant methodology, significant results and conclusion
Reply: Thank you for the comment. Please see lines 8-23.
- The manuscript needs extensive revision for language and grammar especially in experimental sections.
Reply: Thank you for the comment. The manuscript was checked by a native speaker.
- The innovations of this manuscript are limited. Most of the results have already been described in some article with the same topics
Reply: Thank you for the comment. Please see lines 59-72, 140-149 and 448-453.
- the results are not discussed deeply. More discussion and analysis should be included in this article to be published in this journal.
Reply: Thank you for the comment. Please see lines 265-270, 275-283, 285-289, 293-297, 317-321, 326-337, 345-350, 365-376 and 383-388.
Experimental section also should be improved and explain in detail.
Reply: Thank you for the comment. Please see lines 152-157, 196-200 and 225-235 and Figure 1.
Reviewer 6 Report
Clarifications on the points raised below ought to be provided before a definitive proposal regarding publication of the paper is made:
1. Very limited recent research is included in the manuscript and the same has been reflected in the reference section.
2. The majority of the literatures are outdated, and the quantity of literatures cited is likewise insufficient. The study does not contain any recent literatures.
3. The manuscript does not go into considerable detail about how to apply and discuss the findings. The conclusions are similarly lacking in information.
4. Why RAP is replaced only up to 40% Justify the reason.
5. Why VA is replaced by 100% to 60%? Justify the reason.
6. What happens if RAP and VA is added more than specified replacement percentage?
7. What is cost-benefit of your materials as compared to conventional? Why it is not highlighted in your study?
8. Throughout the technical paper, the results are compared and not discussed in a detailed manner with mechanism for improvement in the test results.
Author Response
The authors would like to thank the reviewer for his/her valuable comments. Detailed answers are provided below.
Clarifications on the points raised below ought to be provided before a definitive proposal regarding publication of the paper is made:
1. Very limited recent research is included in the manuscript and the same has been reflected in the reference section.
Reply: Thank you for the comment. Please see lines 104-106, 116-121, 129-149 and reference list.
- The majority of the literatures are outdated, and the quantity of literatures cited is likewise insufficient. The study does not contain any recent literatures.
Reply: Thank you for the comment. Recent references were added. Please see the revised reference list.
- The manuscript does not go into considerable detail about how to apply and discuss the findings. The conclusions are similarly lacking in information.
Reply: Thank you for the comment. Please 265-270, 275-283, 285-289, 293-297, 317-321, 326-337, 345-350, 365-376, 383-388 and 433-447.
- Why RAP is replaced only up to 40% Justify the reason.
Reply: Thank you for the comment. Please see lines 137-139 and 156-157.
- Why VA is replaced by 100% to 60%? Justify the reason.
Reply: Thank you for the comment. Please see lines 137-139 and 156-157.
- What happens if RAP and VA is added more than specified replacement percentage?
Reply: Thank you for the comment. Please see lines 461-466.
- What is cost-benefit of your materials as compared to conventional? Why it is not highlighted in your study?
Reply: Thank you for the comment. Please see lines 36-38 and 467-469.
- Throughout the technical paper, the results are compared and not discussed in a detailed manner with mechanism for improvement in the test results.
Reply: Thank you for the comment. Please see lines 265-270, 275-283, 285-289, 293-297, 317-321, 326-337, 345-350, 365-376 and 383-388.
Reviewer 7 Report
This paper tackles the important worldwide issues of recycling, sustainable development and environmental protection.
Unfortunately, I did not find anything that is a significant contribution to the state-of-the-art. The procedure you applied and most of your findings can be found in many similar papers. I suggest you emphasise how does this paper contribute to previous findings.
The fact that "All blends with RAP develop lower permanent deformations than the mix with 100% VA" (line 355) is questionable, since previous work on this topic suggests otherwise. This is rather contrary to the previous finding regarding the CBR decrease with the increase in RAP content.
Also, there are no remarks on RAP reclamation, RAP particle size distribution, properties and influence of the recovered bitumen.
Therefore, I recommend you reconsider this paper after a major revision.
Author Response
The authors would like to thank the reviewer for his/her valuable comments. Detailed answers are provided below.
This paper tackles the important worldwide issues of recycling, sustainable development and environmental protection.
Unfortunately, I did not find anything that is a significant contribution to the state-of-the-art. The procedure you applied and most of your findings can be found in many similar papers. I suggest you emphasize how does this paper contribute to previous findings.
Reply: Thank you for the comment. Please see lines 59-72, 140-149 and 448-453.
The fact that "All blends with RAP develop lower permanent deformations than the mix with 100% VA" (line 355) is questionable, since previous work on this topic suggests otherwise.
Reply: Thank you for the comment. Please see lines 111-121, 407-412 and references 15, 30 and 35.
This is rather contrary to the previous finding regarding the CBR decrease with the increase in RAP content.
Reply: Thank you for the comment. Please see lines 421-424.
Also, there are no remarks on RAP reclamation, RAP particle size distribution, properties and influence of the recovered bitumen.
Reply: Thank you for the comment. Please see lines 152-157 and Figure 1.
Therefore, I recommend you reconsider this paper after a major revision.
Round 2
Reviewer 1 Report
The comments were all responsed. It is suitable for publication now.
Author Response
The authors would like to thank the reviewer for his/her valuable comments. Detailed answers are provided below.
The comments were all responsed. It is suitable for publication now.
Reply: Thank you for your consideration.
Reviewer 2 Report
The authors have revised the paper However, the below-mentioned major revision remarks must be addressed before the second round of revision.
1. Rewriting the Abstract is necessary. It ought to be succinct and provide readers with information on the background, research question, hypothesis, methodology, key findings, and conclusions of the study that is being presented. Ideally, it should also discuss the main implications and wider context of your findings.
2. Introduction section requires a more detailed discussion leading to this study's problem statement and scope. Also, more literature is needed to be discussed. for instance, in 27-30 please support this statement by citing https://doi.org/10.1007/s10098-022-02347-5; lines 31-32 cite https://doi.org/10.1080/14680629.2022.2064905;line 61 please cite https://doi.org/10.1080/14680629.2021.1995470.
3. The novelty of your work should be apparent and additionally highlighted, together with the objectives of your research, in the last paragraph of the Introduction. Moreover in line
4. The methodology needs to be strengthened. It should be clear and logical so that repeating your work will be possible for interested researchers. If the methodology, or a portion of it, has already been published somewhere else, you should give a brief summary and cite the source.
5. Figures 2-4 and 7 are really unnecessary to show in the manuscript. Proctor test, CBR and triaxial apparatus are not state-of-the-art to be shown in the manuscript also, most of the target readers already know how CBR is being determined therefore Figure 7 is useless here.
6. How did this study cover the sustainability-related aspects for instance take (https://doi.org/10.1007/s11356-021-16912-w; https://doi.org/10.1016/j.enggeo.2022.106899) as a reference to explain sustainability-related aspects and field implications of this study.
7. Lines 281, 265, 267 and 207 need revision, please make sure grammatical and formatting mistakes do not appear in your manuscript. Also, Figure 5 is unnecessary. Line 354 needs references such as https://doi.org/10.1016/j.trgeo.2022.100781; https://doi.org/10.1016/j.jclepro.2022.131345. Check the formatting of your manuscript in detail.
8. The significance of your work needs to be stated more explicitly.
Author Response
The authors would like to thank the reviewer for his/her valuable comments. Detailed answers are provided below.
The authors have revised the paper However, the below-mentioned major revision remarks must be addressed before the second round of revision.
- Rewriting the Abstract is necessary. It ought to be succinct and provide readers with information on the background, research question, hypothesis, methodology, key findings, and conclusions of the study that is being presented. Ideally, it should also discuss the main implications and wider context of your findings.
Reply: Thank you for the comment. The Abstract has been revised. Please see lines 8-28.
- Introduction section requires a more detailed discussion leading to this study's problem statement and scope.
Reply: Thank you for the comment. Please see lines 72-80.
- Also, more literature is needed to be discussed. for instance, in 27-30 please support this statement by citing https://doi.org/10.1007/s10098-022-02347-5; lines 31-32 cite https://doi.org/10.1080/14680629.2022.2064905;line 61 please cite https://doi.org/10.1080/14680629.2021.1995470.
Reply: Thank you for the comment. Please see lines 34-37, 66-68.
- The novelty of your work should be apparent and additionally highlighted, together with the objectives of your research, in the last paragraph of the Introduction. Moreover, in line
Reply: Thank you for the comment. Please see line 81-83, 87-88.
- The methodology needs to be strengthened. It should be clear and logical so that repeating your work will be possible for interested researchers. If the methodology, or a portion of it, has already been published somewhere else, you should give a brief summary and cite the source.
Reply: Thank you for the comment. Please see lines 180-189 and Figure 2.
- Figures 2-4 and 7 are really unnecessary to show in the manuscript. Proctor test, CBR and triaxial apparatus are not state-of-the-art to be shown in the manuscript also, most of the target readers already know how CBR is being determined therefore Figure 7 is useless here.
Reply: Thank you for the comment. The Figures have been removed.
- How did this study cover the sustainability-related aspects for instance take (https://doi.org/10.1007/s11356-021-16912-w; https://doi.org/10.1016/j.enggeo.2022.106899) as a reference to explain sustainability-related aspects and field implications of this study.
Reply: Thank you for the comment. Please see lines 45-47, 172-175.
- Lines 281, 265, 267 and 207 need revision, please make sure grammatical and formatting mistakes do not appear in your manuscript.
Reply: Thank you for the comment. Please see lines 224-225, 280-281 and 296-298.
- Also, Figure 5 is unnecessary.
Reply: Thank you for the comment. The authors feel that this Figure is a good depiction of mechanical performance and it supports the explanation of the experimental process.
- Line 354 needs references such as https://doi.org/10.1016/j.trgeo.2022.100781; https://doi.org/10.1016/j.jclepro.2022.131345.
Reply: Thank you for the comment. Please see lines 367-368.
- Check the formatting of your manuscript in detail.
Reply: Thank you for the comment. It’s done.
- The significance of your work needs to be stated more explicitly.
Reply: Thank for the comment. Please see lines 72-83 and 447-466.
Reviewer 5 Report
- The title is in accord with article
- The manuscript adheres to the journal's standards after revision
- This article contains new aspects, but the authors must underline the major findings of their work and explain how this study represents a progress to other similar published papers. Please provide comparison with other articles
- The Abstract section refers to the study findings, methodologies, discussion as well as conclusion. The Abstract section must be improved. The Abstract should refer to the study findings, methodologies, discussion as well as conclusion. In this form the abstract is too generally
- The keywords permit found article in the current registers or indexes
- In the introduction it is not clearly described the state of the art of the investigated problem. More references are necessary. The references from last years are necessary for demonstrated that this study is actual
- The text can be understood by specialists from other domains
- The paper was written in standard, grammatically correct English, more corrections are necessary
- In Tables are presented necessary results
Author Response
The authors would like to thank the reviewer for his/her valuable comments. Detailed answers are provided below.
- The title is in accord with article
Reply: Thank you for the comment.
- The manuscript adheres to the journal's standards after revision
Reply: Thank you for the comment.
- This article contains new aspects, but the authors must underline the major findings of their work and explain how this study represents a progress to other similar published papers. Please provide comparison with other articles
Reply: Thank you for the comment. Please see lines 72-83.
- The Abstract section refers to the study findings, methodologies, discussion as well as conclusion. The Abstract section must be improved. The Abstract should refer to the study findings, methodologies, discussion as well as conclusion. In this form the abstract is too generally
Reply: Thank you for the comment. The Abstract has been revised. Please see lines 8-28.
- The keywords permit found article in the current registers or indexes
Reply: Thank you for the comment.
- In the introduction it is not clearly described the state of the art of the investigated problem. More references are necessary. The references from last years are necessary for demonstrated that this study is actual
Reply: Thank you for the comment. Please see lines 34-37, 45-47, 66-68 and 172-175.
- The text can be understood by specialists from other domains
Reply: Thank you for the comment.
- The paper was written in standard, grammatically correct English, more corrections are necessary
Reply: Thank you for the comment. It’s done.
- In Tables are presented necessary results
Reply: Thank you for the comment.